# Investigating the Role of Friendship Interventions on the Mental Health Outcomes of Adolescents: A Scoping Review of Range and a Systematic Review of Effectiveness

**DOI:** 10.3390/ijerph20032160

**Published:** 2023-01-25

**Authors:** Tanya Manchanda, Alan Stein, Mina Fazel

**Affiliations:** Department of Psychiatry, University of Oxford, Oxford OX3 7JX, UK

**Keywords:** friendship, mental health, adolescents, intervention, help-seeking

## Abstract

Friendships are crucial in adolescent development. This paper presents a scoping review, followed by a systematic review, to assess friendship interventions and their impacts on the mental health outcomes of adolescents aged 12–24 years. Studies were included if they incorporated a friend or authentic social group in an intervention dedicated to improving mental health outcomes and well-being. Twenty-four studies were included in the scoping review, and eighteen in the systematic review. Data from 12,815 adolescents were analysed; three prominent themes emerged. The most common theme was promoting mental health literacy, followed by supporting help-seeking, and friendship-building/combating isolation. Most evaluations focused on the individual who had received the intervention, rather than their wider friends who would have been potential contacts and experienced any altered interactions. Of the studies focusing on friendship-building, all had positive short-term outcomes but inconclusive long-term effects. Two studies recruited friends from an individual’s authentic social group. While opportunities for improving mental health literacy and help-seeking emerged as key themes, the role of friends in mental health interventions has only been included in a small number of studies. Given that friends are a key point of contact for many adolescents, a better understanding of their domains of influence, particularly on mental health, will potentially enhance interventions.

## 1. Introduction

Adolescence is a particularly formative period in lifespan development and constitutes a significant portion of an individual’s life. The period of adolescence, existing between 10 and 24 years of age, is characterised by social, behavioural, and physical changes, including substantial structural and functional development of the brain [1,2,3]. One of the defining tasks of adolescence is for individuals to learn to navigate complex social relationships, independently make more decisions that can have long-term impacts, establish their own identity, and form intimate relationships [4,5].

Friendships are usually mutually beneficial relationships that individuals voluntarily engage in [6]. Friends can share similar interests, and friendships are often uniquely intimate bonds in an adolescent’s life [6,7]. Previous studies have revealed friendships to be essential to adolescent development [8,9]. Not only do friends become increasingly important during adolescence, but as adolescents start spending more time with their friends, they often prioritise these relationships over others [10]. Typically, the nature of adolescent friendship undergoes significant changes during this developmental period as friendships become more intimate and peer crowd affiliations become more important [8,9,10,11].

However, the developmental changes that individuals experience during adolescence can also put them at higher risk for mental illness, with suicide becoming the fourth leading cause of death among 15–29-year-olds [12]. Depression, anxiety, and behavioural disorders are most prevalent during adolescence, with 50% of those who will have lifelong mental health difficulties starting to experience problems by the age of 14 [12,13]. Adolescent depression is linked to a multitude of psychological difficulties in adulthood. It can limit opportunities at a crucial developmental stage [14], potentially impacting long-term on an individual’s social life, relationships, occupational choices, and socioeconomic status [14].

As friendship takes precedence over a variety of relationships in adolescence, it has the potential to impact a young person’s development and well-being significantly. Research has revealed that adolescents are more likely to turn to their friends, rather than trained professionals or other adults, when in crisis or feeling depressed [15]. Young people also tend to have a higher sense of comfort and trust around their peers than with other contacts in their life [16].

A study measuring friendship quality during early and pre-adolescence found that individuals involved in a ‘mutual friendship’ relation show higher well-being than those without one [17]. Positive experiences with peers or social groups, such as being accepted by a peer group or feeling validated, are associated with a positive evaluation of the self, greater well-being, and fewer internalising problems [18,19]. Friends have also been shown to protect against depressive symptoms during adolescence, with adolescents displaying increased prosocial behaviour toward their friends who feel socially excluded [20,21]. Moreover, they may provide neurobiological advantages, as research shows that friendships contribute to decreased cortisol levels during stressful events [22]. Friendships have been shown to provide adolescents with cognitive and social scaffolding, and adolescents with friends report feeling happier on average and having greater self-esteem [9,23]. Compared to being alone, spending time with peers after experiencing stressors has also been associated with lower feelings of sadness and worry in adolescents [24].

An adolescent’s authentic social group is defined by the social company they keep, namely referring to their friends. Authentic social groups for adolescents can include all of the individuals they consider a friend. They are voluntary social relations with mutual benefits that adolescents engage in as part of their peer-group affiliation.

In the past few years, more studies have emerged on peer-support-based interventions for mental health among youth [25,26]. Researchers have described many types of peer-support programs, such as those classified as “self-help” programs, based on simultaneously helping oneself and others. Self-help is a mutual process without a dichotomy between the helper and the person being helped. Membership in self-help is neither mandated nor charity [27]. Another type of peer-support program is “mutual support,” where individuals or friendship groups voluntarily come together to help each other address common problems and share concerns [28]. Most popularly, previous research on “peer interventions” has focused on an assigned-buddy system to help adolescents with mental health difficulties. This is when an adolescent is matched with a peer, often older, for mentorship, usually with similar mental health experiences.

To our knowledge, there have yet to be any reviews of the breadth of adolescent friendship interventions for mental health. Past reviews have explored peer-mentor interventions and their mental health effects [29], but none have explored interventions involving adolescents’ authentic social groups or friends. This review hopes to address this gap in the literature. The primary purpose of this study is first to use a scoping review approach to identify, map and synthesise the existing literature on friendship interventions and the different roles they might play in supporting adolescent mental health outcomes. Then, we will direct a systematic review to assess the studies where the efficacy of friendship interventions has been evaluated on mental health outcomes in adolescents. The resulting review will help inform the growing evidence base of friendship interventions and mental health outcomes for adolescents.

## 2. Materials and Methods

### Search Strategy and Selection Criteria

As a result of the emergent state of the evidence of using friends and friendships to improve and support adolescent mental health outcomes, a scoping review approach, followed by a systematic review, was chosen as most appropriate for this study. In general, a scoping review is most helpful in examining the extent and range of research in any given area while identifying gaps in the literature [30].

Past peer-based interventions have varied dramatically from each other, and it is unclear how an adolescent’s authentic social group has been formally utilised as a part of these mental health interventions. Therefore, to best synthesise the literature, broad search terms were translated across the following databases in April 2022: EMBASE, Scopus, Web of Science, ERIC, CENTRAL, and MEDLINE. Google Scholar and OpenGrey were further searched for grey literature. For the Google Scholar searches, the results of the first twenty pages were screened by the first author. The first twenty pages included 200 results, and this search strategy was employed based on the probability that there would not be any relevant results after twenty pages [31]. The initial search terms listed in the protocol were further developed with the help of an expert in information technology (EH). Appendix A contain an example search from one of the databases; pilot searches were conducted, and terms were refined to ensure the search returned relevant results. Searches were kept as consistent as possible throughout databases, and MeSH terms and Boolean operators (along with truncations) were used. The search was limited to papers published between 2000 and 2022, and no restriction was placed on language. Forward and backward citation searches were conducted on the papers included in the study, as well as any related papers.

## 3. Scoping Review

### 3.1. Methodology

We conducted our scoping review following the methodological framework proposed for scoping reviews by Arksey and O’Malley and adhered to the Preferred Reporting Items for Systematic Reviews and Meta-Analyses extension for Scoping Reviews (PRISMA-ScR) guidelines [30,32]. The protocol was pre-registered online [33].

The scoping review aimed to identify and map the types of friendship-based mental health interventions in adolescent populations. The current literature uses the terms “friends/friendships” and “peer/peer-groups” interchangeably; however, this review included research conducted primarily using individuals’ authentic social groups. 

The same search strategy was used for both the scoping and systematic review. The scoping review included all relevant papers, regardless of the type and quality of the study. Once the search was completed and duplicates removed, titles and abstracts were screened before full-text review. TM screened titles and abstracts for relevant papers using Rayyan. Studies were coded for acceptance based on the preliminary inclusion/exclusion criteria defined in the protocol. For the scoping review, we decided to remove the criteria of only including studies with a minimum of 10 participants (from the protocol) due to the small number of studies on friendship interventions to try to capture as much information as possible from the available literature.

While adolescence spans between the ages of 10 and 24, our populations of interest were adolescents in secondary school and higher education (defined as 12–24 years). Studies were included if they contained some form of an intervention or a proposed intervention; for those described in multiple publications, the most relevant or most recent publication was included. We excluded studies which addressed peer interventions where the peers in the intervention were not necessarily the adolescents’ friends. However, as the literature uses peer interchangeably with friend, we included “peer intervention” as a synonymous search term and manually excluded studies where the peer was not an authentic friend. Next, we excluded studies which did not have mental health or well-being as one of their outcomes.

We used a descriptive, thematic method to summarise the results. Once the included papers were identified, they were read and screened for mental health themes. We then categorised them into themes based on what each study was targeting, following a similar method to the 2021 review by King and Fazel.

### 3.2. Scoping Review Results

The search resulted in a total of 37,585 articles, of which 37,284 articles were excluded. Following screening the abstracts of the remaining 301 articles, 276 articles were excluded. A final total of 24 studies and grey literature items were included in the review. See Figure 1 for a summary of the search results. The included studies consist of randomised control trials (RCTs), quasi-experimental studies, pre-post studies, protocols, and programmes for adolescent friendship support and interventions. They are summarised in Table 1.

Eight areas focusing on mental health and well-being emerged when examining the included papers (n = 24) (see Table 1). Most of the included interventions involved training adolescents to better recognise signs of mental illness in a friend and provide support or encourage help-seeking. All but one study occurred in a high-income country, and almost all of the studies were in English-speaking countries. More studies took place in Australia (n = 12) than anywhere else, with six taking place in the United States of America, three in the United Kingdom, and one in each of Japan, Switzerland, and India. The high number of Australian studies likely reflects the prominence of Mental Health First Aid (MHFA), and its evaluation, across the country [34].

Both younger adolescents (aged 12–18) and older adolescents (aged 19–24) were included in the studies. Out of the 24 papers, 60% (14) were interventions delivered in person, 32% (8) were interventions exclusively delivered using virtual platforms, and 8% (2) were interventions with both an in-person and remote delivery component.

Figure 2 shows a diagram mapping out the types of studies included in the review and the themes under which they were identified. Three major themes emerged as a result of the review: general mental health and well-being, improving help-seeking, and friendship-building/combating isolation.

#### 3.2.1. General Mental Health and Well-Being

All of the papers, in some way, targeted the improvement and maintenance of general mental health and well-being. However, eight papers were solely focused on general mental health and well-being. A majority of these programmes were delivered using virtual platforms (n = 5) [35,36,37,38,39], with two programmes delivered in person [40,41] and one programme delivered both in person and remotely [42]. Interventions varied from one standalone 50 min session to a programme delivered over six weeks. Some interventions were led by trained teachers in a school, while trained ‘experts’ led others. Two of the included studies were self-directed mental health apps with the aim of equipping adolescents with skills and knowledge to help their friends deal with mental illness. These apps included resources such as videos, which equip adolescents with the skills they need to manage their mental health and support a friend in need. Two studies used and evaluated the MHFA intervention to improve mental health literacy and help-seeking skills among adolescents [36,40]. The dosage varied across studies; for example, one of the remote interventions was delivered using a virtual platform to participants for four hours daily over five consecutive days, by trained peer support experts [37], whereas another consisted of text messages sent to participants multiple times a week for eight weeks [35]. Sessions were informative, and included an aspect of interactivity or an applied demonstration of learned skills—a common theme seen throughout all interventions targeting general health and well-being.

#### 3.2.2. Improving Help-Seeking

Generally, adolescents experience a number of barriers to accessing help as well as concerns about how to provide help to others [16]. Many of the papers we identified worked to address this issue by focusing on improving the help-seeking attitudes of adolescents, either for themselves or for a friend.

Two studies exclusively focused on the mental health of male adolescents and improving help-seeking among them. Both of these studies were delivered in person. One was delivered to a male sports team, and the intervention aimed to increase MH literacy and train adolescents on accessing appropriate resources for themselves or a friend [43]. The intervention was called “Help Out a Mate (HOAM)” and consisted of slideshow presentations, facilitated discussions between participants and presenters, and role-play. Male volunteers with lived experience of mental illness delivered the intervention. The other study implemented a psychoeducational intervention called “Silence is Deadly”, a 45–50 min presentation delivered in secondary schools by trained presenters who act as male role models and included local celebrity athletes as guest presenters [44]. The presentation included videos, informational handouts, and a resource website. It primarily fixated on reducing self-harm in male adolescents by having discussions tailored to the male audience, including strategies on how to help a friend who is struggling mentally, and guiding adolescent males on how to seek help from adults or professionals when needed.

Two studies looked at substance use and mental health. Both used the “Making the Link” intervention, which is a school-based programme specifically developed for teachers to incorporate into their curriculum to facilitate student help-seeking. The intervention consisted of fictional scenarios, videos, and discussions, including a discussion where students discussed the best way to talk to a friend who may need professional help. Out of the two studies using this intervention, one focused on mental health and cannabis use [45] and the other targeted alcohol abuse and depression [46].

There were five studies with the primary aim of suicide prevention amongst adolescents, three were delivered in person [47,48,49], one was delivered remotely [50], and one was delivered both in person and remotely [51]. One of these studies used a teen Mental Health First Aid (tMHFA) intervention, which works to improve general mental health literacy, while also increasing help-seeking intentions, and confidence to provide help to a friend in need.

One of the interventions focused on eating disorders in adolescent girls [52]. This intervention took place during a brief classroom lesson, delivered by the school psychologist. The intervention provided 13-year-olds with the skills necessary to identify and support potential eating disorders in their friends. They were provided with sample language for how to support a friend with an eating disorder, making them aware of the fact that friends turn to each other when under mental distress and the importance of support and guidance towards professional help.

Lastly, two of the included studies aimed to facilitate direct support from friends. One study followed a psychotherapy/peer counselling approach [53]. This online-delivered intervention focused on helping pairs of friends develop skills to support one another in times of distress. The individuals had to practice giving psychological support in a scenario both pre- and post-intervention during a laboratory visit. The second study involved a friend to improve the well-being of adolescents with type 1 diabetes, as well as increase the emotional support offered by their friends [54]. The adolescent with diabetes was asked to bring one of their close friends to an appointment. This friend was educated about their diabetic friend’s condition and given training to help directly support them.

#### 3.2.3. Friendship-Building/Combating Isolation

One study targeted isolation and loneliness [55]. This intervention took place in person and was delivered to all first-year undergraduate engineering students. Students were allocated into a group during a university induction event, where they completed activities to form and strengthen friendship ties with other individuals in the group. The intervention aimed to build friendships in order to combat social isolation and provide a source of social support for mental well-being.

There were two interventions, one in India [56] and one in Australia [57], that were delivered remotely (over the phone) and in person, respectively. Adolescents in these studies were mainly provided with skills to create a social support and friendship network to help their well-being and mental health. In the Indian study, professionals were trained and assigned 10–15 students to offer telephone support to. They would engage in phone calls with the students, listen to their mental distress, offer support using counselling skills, and teach them how they could access support for their distress from their friends and family. The Australian intervention used two programmes in conjunction (Resourceful Adolescent Program and Peer Interpersonal Relatedness programme) to combat depression through friendship-building and friendship support. The interventions were delivered in a school setting and whole classrooms were assigned to receive them.

One intervention delivered in person in Australia addressed social connectedness as a main outcome [58]. This pilot study was delivered by trained leaders, under the supervision of clinical psychologists, to psychologically distressed and socially isolated university students in an effort to improve social networks and connectedness and ultimately improve mental health and well-being.

Throughout all of these interventions, mental health literacy was a prominent theme. All of these interventions were trying to improve mental health literacy in a variety of ways. Some interventions tried to directly improve mental health literacy amongst adolescents [36,49], and most aimed to first improve mental health literacy and then teach participants how to apply the mental health literacy skills by teaching them to support a friend who may be exhibiting symptoms of mental distress.

## 4. Systematic Review

The scoping review allowed us to map the research landscape of existing friendship interventions for adolescent mental health. It illustrated key themes and concepts addressed in the included studies and identified the types of studies which have been conducted. In order to better understand the evidence existing around the effectiveness of the included friendship interventions, a systematic review was then conducted to identify, appraise, and synthesize those studies identified that had available evaluation and post-intervention data (i.e., RCTs; pre-post studies).

### 4.1. Methodology

The PICO (population, intervention, comparator, outcome) approach was employed for this review [59]. The systematic review was completed following the PRISMA 2009 guidelines [60] and registered with Prospero (CRD42022354516).

### 4.2. Inclusion Criteria

The inclusion criteria for the systematic review are described above, as it followed the scoping review with a few additions; it only included published papers that assessed an intervention and were either RCTs, quasi-experimental studies, or pre-post studies. Protocols and grey literature were not eligible for inclusion in the systematic review. There was no limit placed on the number of participants in the study. We included studies that looked at the intervention’s outcomes as well as both qualitative and quantitative studies.

### 4.3. Quality of Included Studies

TM initially filled out the Mixed Methods Appraisal Tool (MMAT) for the studies included. Four studies were then forwarded to MF to evaluate independently; three were chosen at random, and one was selected by TM.

The included studies had some variation in their methodological quality. Most of the included studies were judged to contain possible limitations in at least one criterion (94.4%, 17/18). Almost all of the studies were clear in their description of study participants or the process of recruiting a sample representative of the population of interest (94.4%, 17/18). All but one study [54] included a sample size of 50 or more adolescents, which allowed the authors to warrant conclusions about the efficacy of the interventions. All of the studies targeted mental health outcomes or well-being in some way and had a clear research objective that they addressed in their study.

### 4.4. Systematic Review Results

#### 4.4.1. Description of Studies

A total of 37,585 studies were identified after database searching and screened for relevance. Out of these studies, 18 were eligible for inclusion in the systematic review (see Table 2 for data summarization and Figure 3 for flowchart). The included studies contained an approximate total sample size of 12,815 adolescents. The adolescents ranged from the age of 12.2 to 24.6. Of the included studies, ten were randomized control trials [35,36,37,46,47,50,51,53,55,57], three were pre-post studies [45,48,54], with one study each of a cluster randomised crossover trial [49], a two-arm controlled trial [44], a non-randomized control trial [58], a cluster randomised controlled trial [43], and a quasi-experimental trial [41]. All of the included studies were peer-reviewed published articles and evaluated the short-term and/or long-term effects of an intervention. All of the studies included both friendship-related interventions or outcomes and mental health outcomes. Sample sizes of participants ranged from n = 42 to n = 2456. Included interventions addressed general mental health and well-being, depression, self-harm, male mental health, and isolation/loneliness. All of the interventions included MH and well-being literacy as part of their outcomes, even if the intervention was more focused on, for example, teaching direct support skills. Most of the interventions took place in school or classroom settings, consistent with where many adolescent peer group affiliations occur [8]. Some interventions were delivered exclusively in-person, others were exclusively remote or online and some had both in-person and remote components. Interventions ranged from being delivered in a single 45 min session to a weekly session spanning 11 weeks. As a result of the heterogeneity found between interventions, a meta-analysis was not possible and a narrative synthesis of the results is presented.

#### 4.4.2. Intervention Outcomes

The interventions included in the systematic review had great variation in the outcomes they reported. General mental health and well-being interventions were the most popular, and all studies reported better mental health outcomes and literacy among **friends trained** (FT) in the intervention. Information about **friends receiving** (FR) help from their trained friends was inferred, as it was not directly provided for most of the interventions. Figure 4 depicts an overview of the studies included in the systematic review. Participants were recruited from three different sources in the included studies (schools, universities/colleges, or their communities). The interventions were delivered either remotely, in person, or using a combination of the two. There were two main sets of outcomes for the included interventions, outcomes on FT and inferred outcomes on FR, which are discussed below.

#### 4.4.3. Inferred Outcomes of Intervention on FR

The outcomes of the intervention on FR were not directly measured and were inferred through self-reported data provided in studies by FT. There were three primary inferred outcomes of the intervention on FR. First, the intervention allowed FT to directly provide support to FR. In some studies, such as the one by Hart et al. (2022), the follow-up survey revealed that FT had used their newly learned skills to provide help to a friend in need [49]. Similarly, Craig Rushing et al. (2021) found that adolescents who received the text-message intervention reported helping a friend in need using their acquired knowledge (35). A zoom-based intervention also had FT report that they helped a friend with mental health difficulties following the intervention (37). Next, a common outcome was that FT would encourage FR to seek support for their mental health from an adult or a professional, such as a psychologist or counsellor. Damour et al. (2015) reported that the trained adolescents would encourage their friends whom they noticed suffered from disordered eating to ask for support from a school counsellor [52]. Lastly, FR were helped by FT to reach out directly to an adult or professional to provide help. Damour et al. (2015) also reported this as an outcome of their intervention, where FT reported directly seeking help from an adult for a friend they were concerned about [52]. Similarly, a study employing *MakingTheLink* intervention reported that FT sought help from adults following concerns about a friend [45].

#### 4.4.4. Outcomes of Intervention on FT

There were three major outcomes of the interventions on FT that were measured and reported: FT showed an increase in mental health literacy, knew where to seek support for themselves and others (improved help-seeking), and had increased confidence to support their friends.

### 4.5. Mental Health Literacy

There were 18 interventions included in the systematic review, and 10 of them aimed to improve the mental health literacy of adolescents on some level (refer to Table 2). While these interventions varied in their dosage (one short classroom lesson, to multiple lessons over weeks), their target population (some targeted males only), and their follow-up time, they all showed promising results for increasing mental health literacy among adolescents. One intervention, delivered over the course of five classroom lessons in three weeks, was evaluated through a single-arm trial and concluded that the Youth Aware of Mental Health (YAM) programme was successful in improving mental health literacy and reducing depression in FT as depression severity declined (*p* < 0.001) at post-intervention and six months following the intervention period [48]. In another intervention, delivered in three classroom lessons, tMHFA, showed improvements in adolescents’ mental health literacy where, 12 months after receiving the intervention, trained adolescents rated adults as helpful sources for mental health aid [49]. A shorter 45 min male-targeted intervention, Help Out a Mate (HOAM), also reported increases in knowledge about depression and anxiety for trained adolescents, with the knowledge being retained for four weeks post-intervention [43].

#### 4.5.1. Improving Help-Seeking 

Changes or increases in help-seeking attitudes toward friends, along with the confidence to support a friend in emotional distress or going through a mental health illness, were reported in many studies. One trial evaluated the quality of mental health first aid intention provided by adolescents who had received the teen Mental Health First Aid (tMHFA) intervention. The study reported that participants receiving the intervention provided better quality support to hypothetical friends in a fictional scenario with a mental illness [49]. The primary purpose of this intervention was to improve the knowledge, attitudes, and behaviours of adolescents so that they can better support their friend(s) in a mental health crisis. This study also found that confidence to help a friend significantly improved more after the tMHFA intervention, and was retained for 12 months after the intervention, when compared to the control condition. A few of the included interventions employed MHFA [36,40,49], and all yielded positive results. Consequently, there is reasonable evidence to support the use of MHFA programmes, which increase MH literacy in adolescent friendship interventions while equipping them with confidence to seek help and provide support.

Similarly, a study employing a student help-seeking programme called “Making the Link” found that students after the intervention were more likely to seek help for themselves and a friend and were significantly more informed about mental health and substance abuse disorders [45]. This was further supported by another randomized control trial which found similar increases in help-seeking behaviour when employing the same intervention for a different substance abuse disorder [46]. The intervention included a shorter training time than the one by Hart et al. (2022) at two brief sessions, in the hope that “Making the Link” could be a scalable intervention incorporated into the school curriculum.

The Youth Aware of Mental Health (YAM) programme was successful in increasing the help-seeking intentions of FT towards their friends (*p* < 0.001) at post-intervention and six months following the intervention period [48].

One study used a pre-post design to evaluate the individuals’ skills to support their friends with diabetes [54]. Diabetic adolescents were asked to bring a best friend with them to the intervention. The intervention focused on improving the quality of direct support provided by FT and was successful as it increased individuals’ knowledge about their friend’s condition and taught them ways in which they could offer support. Following the intervention, FT provided a significantly higher amount of support to their diabetic friends [54]. However, no similar interventions were identified in the literature.

Similarly, the web-based intervention in the study by Bernecker et al. (2020) asked participants to bring an individual from their existing social circle for participation. This was the only peer-counselling course intervention found in the literature, and it evaluated whether peers can be trained to provide non-professional psychological support to one another. The study found significant results where participants who took the web-based counselling course were better listeners and better at offering support to their friends [53].

A 2015 study by Damour et al. worked on improving the help-seeking attitudes of FT toward their friends displaying signs of an eating disorder [52]. After the intervention, FT were more likely to seek help from an adult for a friend at risk of an eating disorder. The likelihood of reporting concerns about the disordered eating of friends was measured at baseline, immediately following the intervention, and approximately three months later. Variables (the willingness to talk to a friend about eating behaviours, encouraging a friend to talk to an adult, talking directly to an adult about a friend’s disordered eating) were measured at each interval and comparisons were made between the eighth and ninth graders. The results suggest that intervention programmes such as this one can be implemented successfully in high schools and might be advantageous for reducing or managing disordered eating behaviours and the risks associated with them [52].

#### 4.5.2. Friendship Building

The three studies that had interventions aimed at friendship building as a primary outcome had different findings at long-term follow-up. One of these interventions looking at younger adolescents (12.2 years old) used two different kinds of interventions concurrently (RAP and PIR) to improve adolescent social networks and promote depression literacy, help-seeking, and teach support skills [57]. RAP incorporates cognitive behavioural therapy and interpersonal psychotherapy principles to teach adolescents how to think resourcefully, problem solve, identify and access support networks, etc. PIR teaches adolescents about the importance of friendships, what peers look for in a friend, social skills training, resolving conflicts with a friend, etc. The combination of these two interventions significantly reduced depressive symptoms, while significantly increasing social functioning with friends relative to their friends in the other conditions [57]. Another friendship-building intervention looked at older adolescents (first-year undergraduates) and, in order to facilitate friendships, students were assigned to groups for activities. The study evaluated both short-term and long-term effects of the intervention and found that although students made friendships through which they sought emotional support in the short-term, these friendships had diminished by the one-year follow-up [55]. Lastly, one of the interventions, Groups for Health (G4H), recruited undergraduate students and included those who screened positive for psychological distress and social isolation [58]. The G4H intervention tried to increase social connectedness and group identity formation in an effort to help participants build friendships to combat social isolation and psychological distress. The intervention successfully improved mental health, well-being, and social connectedness, both immediately after the intervention and at a six monthly follow-up. Furthermore, the participants maintained their social group identification at follow-up [58].

Only one intervention (Kognito Face2Face) incorporated the use of simulations and virtual peers to train adolescents in how to support their friends with a mental illness or in times of emotional distress [50]. The intervention was successful in eliciting supportive responses from participants, and participants were more likely to support a friend in times of mental distress after the intervention [50].

## 5. Discussion

There are compelling reasons to focus on friendships for adolescent mental health interventions given the prominence of these relationships for this age group. The studies included in this scoping review and systematic review mainly targeted three broad and overlapping areas: general mental health and well-being; help-seeking attitudes for the person in the intervention or their friends; and, finally, friendship building and combating social isolation. The studies identified in both the scoping and systematic reviews reported here, highlight how, especially in the area of mental health literacy, there is good evidence that this is likely to be helpful, especially for the friend being trained in the intervention (FT) and possibly for the **friend receiving** (FR) the intervention.

In the scoping review of 24 studies, almost all the interventions focused on how friends being trained (FT) can support members of their own social circle who may present with mental health difficulties or distress. Only two of the studies reported the effects of the intervention on the **friend receiving** (FR) mental health support [53,54]. A broad range of different interventions were tried, covering those improving general mental health literacy to male-only interventions targeting the reduction of self-harm. Some of the interventions, such as MHFA and “Making the Link”, have been well studied with consistent positive findings [40,45,46,49]. The majority of other interventions are novel, or at least novel in their implementation as an adolescent friendship intervention. Of note, most of the interventions have been studied and developed in Australia, with the majority of the rest in other high-resource settings.

The systematic review assessed the efficacy and feasibility of friendship interventions for adolescent mental health. The included RCTs mostly evaluated the effects of the interventions on FT. There was a total of 12,815 adolescents included in the 18 studies identified in the systematic review. The mental health first aid interventions were most promising for increasing mental health literacy among adolescents, and also the ones used most commonly. Long-term effects of the interventions on either FT or FR were not assessed—specifically, long-term effects on mental health and well-being. However, all of the included studies reported some positive effects of the intervention on adolescents’ mental health and well-being, especially immediately after the intervention.

The first onset of major depressive disorder can, for many, occur around the time of early adolescence [61]. As adolescents are more vulnerable to mental health disorders, it is imperative to explore prevention and intervention methods to reach larger populations. Past research has demonstrated that friendships during adolescence can reduce depressive symptoms, especially in high-risk populations who can be harder to reach by mental health professionals [62]. This review aimed to map and evaluate the types of friendship interventions which exist and their role in adolescent mental health outcomes in an effort to understand the role of informal mental health supports, such as friends.

Of the relatively small number of friendship mental health interventions that have been studied, most vary in their training time, trainer qualifications, and delivery (being in person and/or online), making comparisons difficult. Interventions delivered in-person, as well as delivered remotely, have both had positive outcomes. For example, a zoom-delivered intervention [37] reported that FT utilised skills to help a friend with mental illness, and a text-message-based intervention also reported FT using their newly learned skills to help a friend [35]. Along the same lines, interventions delivered in-person also produced positive outcomes such as increased mental health literacy and increased ability to provide support.

The shortest intervention, delivered in a single 45–60 min in-person session, reported improved help-seeking intentions for FT but did not find an improvement in confidence to support a friend [44]. This may suggest that a single session may not be enough to train adolescents to support their friends or increase their perceived ability to help their friend. However, another 45 min single-session intervention reported that the intervention increased intentions to provide help to a friend, but no improvement was observed for help-seeking attitudes or intentions [43]. While both of these studies were in-person interventions, male-targeted, and short in length, it is unclear why one improved help-seeking intentions and one did not, or why one had no effect on confidence to provide help to a friend and one increased intentions to provide help to a friend.

Sample characteristics in the studies were equally varied and included students attending public or private schools as well as university students from less deprived backgrounds. Craig Rushing et al. (2021) included an entire sample from a less-represented group and Aseltine and DeMartino (2004) included a partial sample of students from a disadvantaged neighbourhood [35,47]. The results, therefore, need to take into account potential differences in these samples; it might be that adolescents who face adversity may be at higher risk of mental health disorders, and as a result may potentially benefit more from such interventions [62]. Furthermore, adolescents from disadvantaged backgrounds may turn to informal sources of support more readily as there might be limited provision in their areas or they might have delayed access to formal supports such as those offered by mental health services.

There were few interventions that directly targeted an adolescent’s social group. Most of the interventions indirectly targeted adolescents’ friends by teaching MH support skills and increasing MH literacy so that adolescents could better support their friends who may be going through mental distress in the future. Only two studies brought in members from an adolescent’s authentic social group to teach them how to support their friends [53,57]. Of note, both these studies reported successful outcomes and worked to promote support skills. One intervention was delivered entirely online and to an older adolescent population over the course of four weeks [53], and the other was delivered entirely in person to a younger adolescent population over 11 weeks [57]. Moreover, none of the interventions had any long-term follow-up data; the longest was a 12-month follow-up for a befriending intervention and did not assess mental health and well-being at the follow-up [55]. Some interventions were delivered once for less than an hour, while some were conducted over several weeks. Almost all of the interventions included an element of interactivity during the training process—this manifested in the form of role-plays, group/partner discussions, and fictional scenarios, for example. Making the programme interactive might be important to encourage engagement amongst adolescent participants.

Figure 2 maps out the range of the interventions included in the review. While the interventions covered different concerns, there were many gaps identified through this review. First, the interventions lacked information about the direct outcomes on FR, with minimal information providing evaluations of friend helpfulness on FR. Second, the included studies did not identify the types of friends that adolescents may turn to, for example, understanding if the friends they reach out to are primarily in-person or online. This is an important area to clarify as the online era has allowed adolescents to keep in touch with their friends, as well as build their friendships, through different platforms, such as social media, with the classification of friendships becoming more sophisticated [63]. Recent literature has found online social platforms to be beneficial for adolescent friendships, with younger adolescents with access to online platforms at times using online and offline communication with little differentiation [64]. Apart from social media, communication through and during online gaming is also a popular way of online communication and friendship-building among adolescents, as spending time working together on the same task can strengthen friendships [64]. Social media can also serve as a supportive platform, as active adolescent users of social media report seeking informal support from friends and peers online, which is why it is important to introduce interventions which involve online friends and understand whether these interventions can play a role in enhancing online friendships [65]. Third, none of the studies looked at interventions to help adolescents support each other in times of grief, trauma, or loss, even though research suggests that emotional support is the most desired type of support following a traumatic loss and adolescents desire increased support during these time from their friends [66,67]. Fourth, the interventions included here did not employ some important evidence-based active ingredients, such as behavioural activation, which works by increasing engagement with positive activities, and is proven to be successful in reducing depression for young people [68]. Another active ingredient not employed or evaluated by the included studies was self-evaluation. Studies highlight the importance of positive self-evaluation as an active ingredient in the treatment of adolescent depression [69]. Finally, the included studies did not investigate social inclusion as a component of their intervention, an important area to better understand. Interventions should have explored educating adolescents about the poor mental health outcomes of being socially excluded, as research shows that some adolescents are prosocial and empathetic towards their socially excluded peers [21]. Adolescents struggling with mental illness can also often feel ostracised among their friend groups, and so finding ways to reduce that might be needed to enable their friends to become more socially included with their larger peer group.

Based on the findings of this review, it is evident that friendship interventions to support adolescent mental health are sparse. Furthermore, the risks of friendship interventions need to be better understood and addressed in future studies. As some of the studies are implemented in schools, and some are implemented privately, as a part of a sports team, or in treatment for individuals, it is unclear what the best approach would be for an effective intervention. Friendship interventions that worked to improve mental health literacy reported positive outcomes, and future interventions should consider continuing to incorporate mental health literacy as part of their programmes. Based on the themes of the interventions included in the scoping review (see Table 1), it is evident that there are many targets for adolescent friendship interventions. However, as some mental health and well-being outcomes have only been targeted by a single intervention, there is not enough evidence to draw conclusions.

Further, the effects of the intervention on FR (as seen in Figure 4) should be evaluated in greater detail to understand if the intervention is being successfully delivered to FT and encouraging them to provide appropriate and helpful support. No intervention directly measured the quality or helpfulness of the support received by FR. This sheds light on a substantial gap in the literature to ensure the intervention benefits the target individuals.

## 6. Strengths and Limitations

This review has a number of strengths. It is the first of its kind to comprehensively investigate the literature on friendship interventions and their role on adolescent mental health outcomes. The review was widespread, as it included multiple large databases and expansive search terms to try and capture all the interventions, including friends/friendships. The scoping study in the first instance also allowed us to capture and report grey literature that adds to the overall representation of friendship interventions in mental health research. Organising the papers by theme allowed us to paint a clearer picture of the types of mental health outcomes targeted by friendship interventions for adolescents. Systematically reviewing the interventions published and tested allowed us to highlight the gaps in the literature, such as the lack of studies evaluating FR and the lack of long-term outcomes for interventions. This review also highlighted the potential for friendship interventions as a valuable tool to help provide support to adolescents who may not immediately have access to formal sources of mental health support.

This review has several limitations. Firstly, although the search criteria were comprehensive, it may not have captured all of the papers involving friendship interventions. Friendship interventions are sparse and remain largely unexplored. The terminology used in papers which employ friendship interventions is inconsistent. As a result, some papers using unconventional terminology may have been missed. However, due to the nature of the literature around friendship interventions, search terms had to be broad to capture relevant results.

Secondly, most interventions’ mental health and well-being outcomes were not empirically measured using a valid and reliable scale or measure of mental health. This may have led to inconclusive or undetermined changes in mental health and well-being as a result of the intervention, and could not allow us to effectively determine the impact of adolescent friendship interventions on adolescent mental health. While broad age ranges of adolescents were represented in the review, it is still unclear which of the included interventions, if any, can be used interchangeably for younger or older populations of adolescents. The interventions also mostly focused on self-help or improving mental health literacy. The focus was rarely concentrated on solely providing support to friends, as this was often a secondary or tertiary outcome. Next, as long-term outcomes of the interventions were not evaluated, it is unclear whether booster doses of the intervention are needed periodically to revive adolescents’ knowledge and ability to provide helpful support to their friends. Another notable limitation is the delivery of the intervention programmes. Some interventions used trained staff to disseminate the programme, whereas others trained teachers in a school and others recruited youth volunteers with lived experience of mental illness. This may have led to biases in intervention presentation, as some trained leaders may have had some pre-existing relationship with the students (i.e., teachers). As a result, specific interventions may show inconsistencies in effectiveness depending upon the individual delivering them to the adolescent. Lastly, the included interventions, coming mainly from high-income countries, had limited adaptation for different socio-cultural contexts. Friendship dynamics and values can differ across cultures [70], and future studies should delineate which type of interpersonal interaction is being targeted if working in a new environment. Identifying components that can be tailored, within interventions, to different contexts will likely better enable interventions to be delivered in new areas. The many apparent gaps in intervention research on mental health will hopefully serve to encourage the development of friendship interventions for adolescent mental health support in the future.

## 7. Conclusions

This scoping and systematic review aimed to collect and assess the literature on friendship-based interventions and their role in adolescent mental health. A comprehensive search was carried out on multiple databases to ensure a broad range of the literature was captured. While the search yielded many results, the papers eligible for inclusion were few. Most of the included studies aimed to improve mental health literacy amongst adolescents to better equip them to support themselves and their friends. Help-seeking attitude improvement was also a notable theme. Most of the interventions worked to either improve social support, teach individuals about the signs of mental illness in their friend, or improve confidence to support a friend. The interventions mainly focused on the **friends trained** (FT) and their outcomes.

Friends are seldom involved in mental health interventions, despite being a key source of informal support for adolescents, especially at times of crisis [15,71]. The importance of friends for many adolescents, and the support they provide to each other during adolescent development, is indisputable. This review sheds light on the friend interventions studied to date. Friendship interventions have the potential to improve and protect adolescent mental health. They can be implemented in a natural setting, using the adolescents’ pre-existing social circle as a source of informal support. The results highlight how little is known about this important potential area of mental health intervention, and developing a multifaceted lens of enquiry into whether and which supports might best be delivered by which friends, might enable more adolescents to access support in their times of need.

## Figures and Tables

**Figure 1 ijerph-20-02160-f001:**
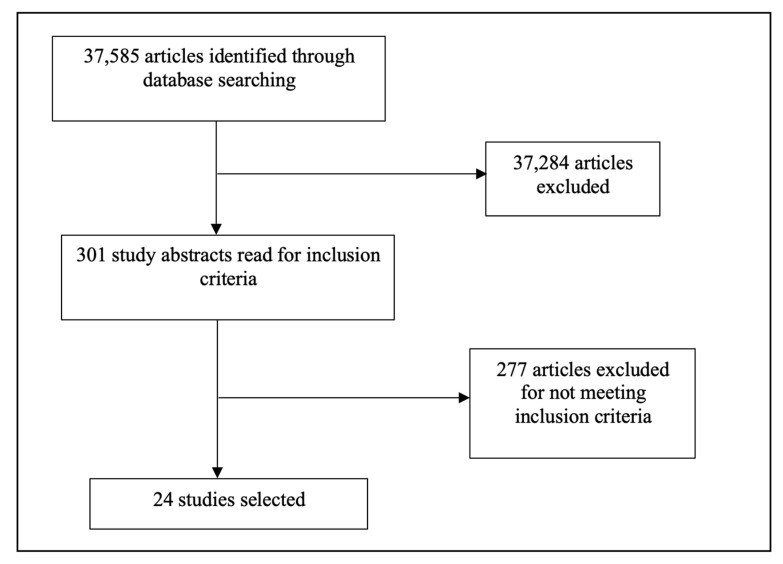
Flowchart depicting a summary of searches and results for scoping review.

**Figure 2 ijerph-20-02160-f002:**
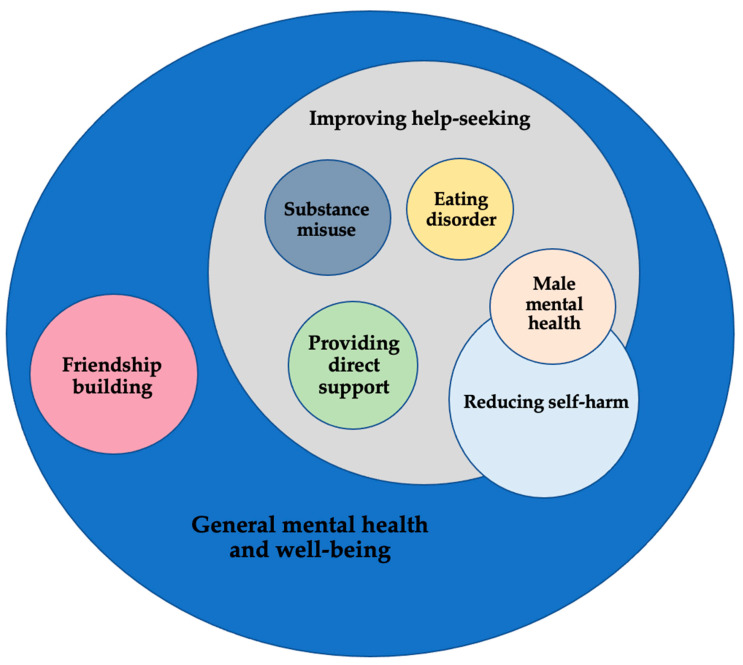
Mapping targets of the included friendship interventions.

**Figure 3 ijerph-20-02160-f003:**
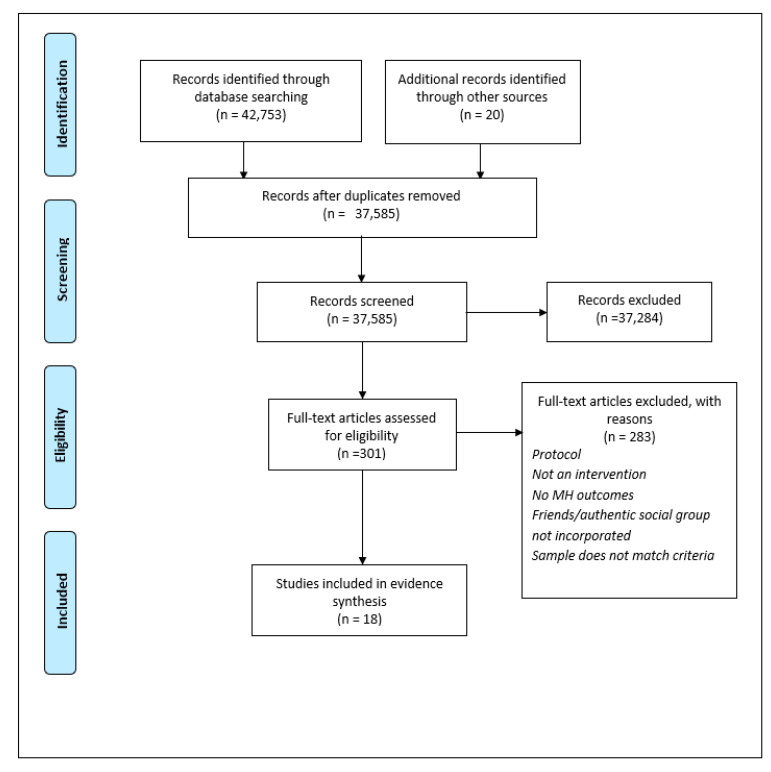
Flowchart depicting summary of searches and results for systematic review [60].

**Figure 4 ijerph-20-02160-f004:**
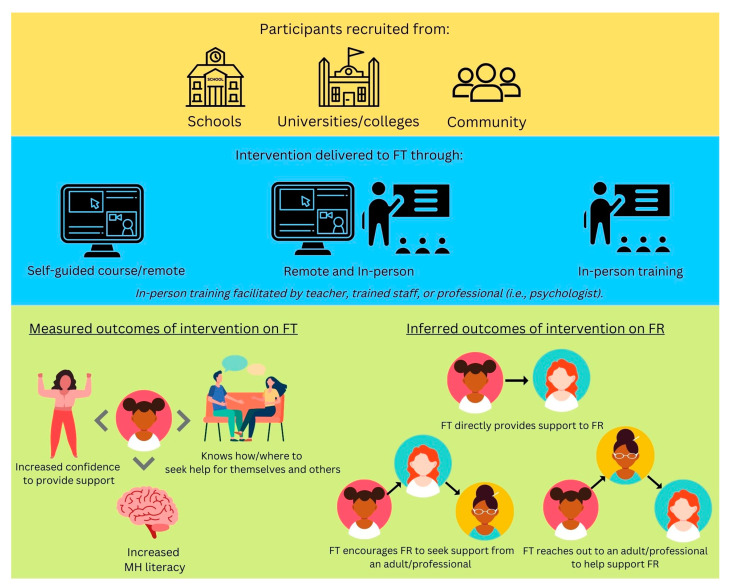
An overview of the studies evaluated in the systematic review, illustrating the typical implementation and results of the included interventions. Note: FT denotes **friends trained** in the intervention. FR denotes **friends receiving** support. MH denotes mental health.

**Table 1 ijerph-20-02160-t001:** Characteristics of included items for scoping review organised by theme.

Author and Year	Location	Average Age	In-Person (IPR) or Remote (RMT)	Intervention Characteristics
**Eating Disorder**				
Damour et al. (2015)	United States of America	13.9–14.9	IPR	Eighth and ninth-grade class lessons to females on identifying a potential eating disorder in themselves or their friends. The intervention aimed to teach them to approach an adult if they suspect a friend has an eating disorder and how to support them.
**Friendship-building/Social Connectedness**				
Boda et al. (2020)	Switzerland	First year undergraduates	IPR	First-year undergraduate engineering students were randomly assigned to groups for a few hours on a student information day to facilitate friendships, especially mixed-gender friendships, to combat feelings of isolation and increase well-being and support.
Devassy et al. (2021)	India	18–35	RMT	A telephone-based intervention to encourage participants to seek support from their social circle (friends, family, significant others) to combat depressive symptoms, anxiety, etc.
Haslam et al. (2016)	Australia	20.2	IPR	Pilot study of an intervention for university students to improve students’ well-being, mental health, and social connectedness by helping participants build a social support network of friends and social group identity through a group-based intervention.
Rose et al. (2014)	Australia	12.22	IPR	Whole classrooms of adolescents were taught skills to make friendships in the first intervention, followed by a second intervention in conjunction to reduce depressive symptoms and increase general well-being, while promoting support skills for friends.
**General mental health and well-being**				
Ashoorian et al. (2018)	Australia	24	IPR	Mental Health First Aid (MHFA) intervention for university student groups that are well-placed to provide early intervention to their peers (ex. student representatives, student leaders, etc.) by teaching them skills to recognize signs of mental distress/illness. Students were trained to apply MHFA skills to their friends.
Birrell et al. (2022)	Australia	15.9	IPR and RMT	A smartphone peer support app and introductory classroom lesson aimed at empowering adolescents to access evidence-based information and tools to better support peers regarding anxiety, depression, and substance use-related issues. The programme contains external links for adolescents to access if they are concerned about a friend or themselves.
The Check-In App. BeyondBlue (mHealth App)	Australia	Young people	RMT	A smartphone app co-produced by young people to facilitate talking to a friend who might be struggling with mental illness. The app provides the user with advice for how to help their friend, while also looking after their own mental health. The interface also provides online and phone resources.
ConNetica. Chats for life (mHealth App)	Australia	Young people	RMT	A smartphone app co-designed with young people to help them plan a conversation with a friend who may be struggling with a mental health issue. The app provides video guides for users for how they can support their friends’ mental health and well-being.
Craig Rushing et al. (2021)	United States of America	15–24	RMT	BRAVE is a text message-based intervention for Native youth to promote mental well-being and help-seeking skills. Text messages and role-model videos are sent to Native youth participating in the study in an effort to increase social support, help-seeking, mental health literacy, and well-being.
Davies et al. (2018)	United Kingdom	19.9	RMT	Mental health literacy and first aid training using MHFA intervention for medical students to teach them how to recognize signs of mental illness, and increase confidence in providing mental health support to friends.
Pavarini et al. (2022)	United Kingdom	16–18	RMT	A training programme with the aim of teaching adolescents how to best support their friends’ mental well-being during the COVID-19 pandemic. The programme was delivered to adolescents by a trained team of Youth Era peer support experts over five consecutive days.
Yamaguchi et al. (2020)	Japan	15–16	IPR	Short Mental Health Program (SMHLP) is a 50 min teacher-led intervention to teach adolescents about mental health symptoms, improve help-seeking behaviours and increase intentions to support a friend displaying signs of mental distress.
**Male mental health**				
Calear et al. (2021)	Australia	16–18	IPR	An intervention to promote help-seeking for emotional distress and self-harm amongst adolescent males. The intervention aimed to encourage adolescent males to seek help from and provide support to friends in times of distress or suicide risk.
Liddle et al. (2021)	Australia	14.3	IPR	A 45 min workshop, Help Out a Mate (HOAM), designed to educate male adolescents on mental health, support skills, and how to access appropriate resources for themselves or a friend. The intervention was delivered in a sports setting and included a PowerPoint presentation, facilitated discussions between participants and presenters, and brief role-plays.
**Psychotherapy/Peer Counselling**				
Bernecker et al. (2020)	United States of America	24.6	RMT	An intervention called “Crowdsourcing mental health (CMH)”, a self-guided web-based course, teaches reciprocal supportive psychotherapy skills to pairs of friends. Dyads alternate between helper and receiver role.
**Reduction of self-harm**				
Aseltine and DeMartino (2004)	United States of America	Grades 9–12	IPR	Signs of Suicide (SOS) school-based prevention programme to teach adolescents how to recognize the signs of depression and/or self-harm and empower them to support and intervene when a friend may be exhibiting these signs.
Coleman et al. (2019)	United States of America	20.5	RMT	An online suicide-prevention gatekeeper training intervention called “Kognito Face2Face” was given to enhance adolescents’ help-seeking attitudes and peer-help attitudes. The trainee adolescent is put in a simulated college social environment in Kognito and interacts with virtual peers. The trainee has to identify peers who may be at risk by engaging in dialogue, deciding if a professional referral is needed, and referring their peer to a professional for help.
Hart et al. (2022)	Australia	15.87	IPR	Teen Mental Health First Aid (tMHFA) is a mental health literacy programme implemented in secondary schools in a classroom setting by trained professionals to teach adolescents how to respond to a friend at risk of self-harm. Students are trained to recognize signs of mental distress and suicidality in their peers. Training involves vignettes of virtual peers in distress. Changes in providing support are measured through a survey administered before training and a survey administered 12 months after training.
McGillivray et al. (2021)	Australia	14.4	IPR	Implementation of a universal mental health programme, Youth Aware of Mental Health (YAM) to reduce suicidal ideation and depression, and increase help-seeking. YAM includes discussions and role-plays based on the following topics: What is mental health, self-help advice, stress and crisis, depression and suicidal thoughts, helping a friend in need, and getting advice: who to contact.
Wyman et al. (2010)	United States of America	15–16	IPR and RMT	Schoolwide suicide prevention intervention (Sources of Strength) delivered through trained peer leaders to encourage their friends to seek help for themselves and their friends at risk of suicide.
**Substance misuse**				
Berridge et al. (2011)	Australia	15	IPR	A school-based health promotion programme (*MakingTheLink)* that promotes help-seeking behaviour for mental health issues and cannabis use among young people. Scenarios are used to train students in how to respond if a friend shows similar signs.
Lubman et al. (2017)	Australia	14.9	IPR	A school-based health promotion programme (*MakingTheLink)* where secondary school students were presented with two vignettes of peers, Sarah and Samuel, depicting depression and alcohol misuse to improve help-seeking intentions and encourage adolescents to support their friends.
**Well-being related to other illnesses**				
Greco et al. (2001)	United States of America	13.1–13.6	IPR	Diabetic adolescents and one of their friends participated in an intervention to improve knowledge about diabetes, social support, and social functioning. The intervention aimed to increase positive peer involvement and emotional support for friends with diabetes.

**Table 2 ijerph-20-02160-t002:** Characteristics of included studies for systematic review.

Author, Year, and Location	Research Design	N	Approximate Average Participant Age	Intervention Type	Frequency/Duration of Programme	Participant and Peer Selection Process	Training	Intervention Outcomes and Findings
Aseltine and DeMartino (2004), USA [47]	RCT	2100	14–18 *	Mental illness prevention intervention	2 days	Whole classes recruited from secondary schools.	Signs of Suicide (SOS) is a school-based prevention programme delivered in a classroom to help students identify markers of suicide in themselves and their friends. Schools receive a kit of materials containing the DVD of informational videos, discussion guide, screening forms, and other educational and promotional items. They also receive the procedure manual that describes how to implement the programme.	Short-term effects on students’ behaviours were observed with more adaptive attitudes towards depression and suicide. Students were more likely to provide support to friends showing signs of distress. Discussions conducted in classes several months after exposure to the programme revealed that students were unlikely to seek out school staff for MH concerns, primarily because of confidentiality. Instead, students reported that friends were the first people they would turn to when feeling depressed.
Bernecker et al. (2020), USA [53]	RCT	60	24.6	MH literacy and support skills intervention	Course taken over four weeks.	Recruited participants asked to recruit a friend from their existing social circle.	Participants were brought into the laboratory to first talk about their stressors before the course. Then, participants completed an online training course and had to use skills learned to go through helper and talker role in vignettes.	Participants in the intervention group changed some behaviours to better support peers (e.g., talking less about themselves and listening more when a friend shares a problem). No change in feelings of supportiveness or closeness with their friends was reported.
Berridge et al. (2011), Australia [45]	Pre-post	182	15	MH literacy and help-seeking intervention	Two class lessons.	Students recruited from Year 10 (aged 15) classes.	A school-based intervention where facilitators with backgrounds in teaching and mental health deliver the programme in classrooms.	Students and teachers reported feeling better informed about mental health and substance abuse disorders. Students who completed the programme reported increased confidence to seek help for themselves or a friend.
Boda et al. (2020), Switzerland [55]	RCT	226	18–19 *	Friendship building and MH support	Range	An incoming cohort of engineering students recruited	Students were randomly assigned to groups during their orientation week and completed activities to facilitate friendship.	Students in the intervention group developed more friendships and had more individuals they could turn to for emotional support. A 12-month follow up revealed that friendships did not last.
Calear et al. (2021), Australia [44]	Two-arm controlled trial	594 (males only)	16–18	MH help-seeking (male targeted)	45–60 min presentation	Government colleges and private secondary schools invited to participate.	Menslink ‘Silence is Deadly’ programme is a psychoeducational intervention including key statistics about mental health presented alongside personal experiences of the presenters. It includes a presentation, supporting website, videos, and wristband.	Intervention significantly increased help-seeking intentions of participants from friends for emotional problems and mental distress. No difference was found for confidence to support their friends or reduce MH stigma.
Coleman et al. (2019), USA [50]	RCT	69	20.5	Suicide prevention and gate-keeper training	Range	Recruited from two large undergraduate lecture courses.	Participants were given avatar-based online training where they were trained in identifying and clarifying risk and then in encouraging an at-risk friend to seek help.	The intervention was successful in increasing participants’ intention to provide support to a friend with mental illness and increased chances of referring a friend to professional services (i.e., counselling centre).
Craig Rushing et al. (2021), USA [35]	RCT	833	15–24	MH literacy and help-seeking	8 weeks	Social media and SMS recruitment of Native American and Alaska Native teenagers from the community.	SMS text messages about wellness, MH, help-seeking, etc., were sent out to participants three times per week for eight weeks.	Participants reported using the informative SMS text messages to help family and friends. Follow up after 3 months revealed 22.4% of participants reported using skills learned from SMS messages to offer help and by 8 months there was an increase to 54.6% of participants.
Davies et al. (2018), UK [36]	RCT	55	19.9	MH literacy and support skills	6 weeks	Medical students recruited.	MHFA E-learning course	Intentions and confidence to help a friend with mental illness increased and stigma towards mental illness decreased. An increase in MH first aid skills was also found.
Greco et al. (2001), USA [54]	Pre-post	42	13.1–13.6	Integrating friends into care	4 weeks	Diabetic adolescents asked to bring a ‘best friend’.	Four 2 h education and support group sessions led by licensed psychologists.	Friend support increased following the intervention. Trained ‘friends’ were more educated on their friend’s condition and learned how to help support the well-being of a diabetic friend.
Hart et al. (2022), Australia [49]	Cluster randomised crossover trial	1605	15.87	MH literacy and peer support training for suicide prevention	Training for the tMHFA, three 75 min classroom sessions	Students recruited from high schools.	MH literacy intervention provided to adolescents, with activities and vignettes led by an external instructor.	Students receiving tMHFA training more likely to report improved recognition of suicidality and appropriately respond to and provide first aid intentions towards a peer at risk of self-harm than students in active control arm.
Haslam et al. (2016), Australia [58]	Non-randomized control trial	51	20.2–20.95	Friendship building and isolation	Modules of 60–75 min once a week over four weeks. Last module delivered a month after the 4th.	University undergraduates and students from a concurrent study (control). All participants screened, and only those with some kind of psychological distress or social isolation included.	Five-module psychological intervention targeting the development and maintenance of social group relationships to improve psychological distress arising from social isolation.	Groups 4 Health (G4H) intervention was successful in increasing mental health outcomes, well-being, and social connectedness, both on programme completion and 6-month follow up. Participants also had improved social connectedness with peers and increased group-identification.
Liddle et al. (2021), Australia [43]	Cluster randomised controlled trial	102 (males only)	14.3	MH literacy and help-seeking intervention	45 min session.	Recruited from a male community football club.	Presentation, facilitated discussions and brief role-plays delivered in a sports context. Adolescents were provided with a card listing support and resource options. Volunteer student facilitators with lived experience of mental illness led the workshop.	Compared to the control group, the adolescents receiving the intervention showed improvements in mental health literacy for anxiety and depression, along with intentions to provide help to a friend. Further, improvements in attitudes that promote help-seeking and reduce stigma were also observed.
Lubman et al. (2017), Australia [46]	RCT	2456	14.9	MH literacy and help-seeking intervention	Range	Participants recruited from secondary schools.	*MakingTheLink* is a school-based intervention where participants are presented with vignettes (depression and alcohol misuse) and asked to respond to scenarios to provide help.	Pre-post survey results revealed that, after the intervention, adolescents were more likely to seek help for friends showing signs of depression or alcohol misuse. They were also more likely to seek help for themselves.
McGillivray et al. (2021), Australia [48]	Pre-post	556	14.4	MH literacy and suicide prevention	Five classroom lessons over three weeks.	Secondary schools invited to participate.	The Youth Aware of Mental Health (YAM) programme is a school-based programme delivered by trained facilitators and includes booklets, discussions, and role-play activities which allow adolescents to discuss mental health and improve problem-solving and emotional functioning in difficult real-life situations.	There was a reduction in depression severity and suicidal ideation, and an increase in help-seeking intentions at 3 months post-intervention and 6 months follow-up.
Pavarini et al. (2022), UK [37]	RCT	100	16–18	MH literacy and support skills intervention	5 days	Poster/advert community recruitment.	Uplift peer support training course delivered online (Zoom). Interactive sessions including sharing and hands-on activities were delivered through breakout rooms or WhatsApp.	All participants reported an increase in MH knowledge and reported using the learned skills to support at least one friend/peer in their social circle with mental health difficulties.
Rose et al. (2014), Australia [57]	RCT	210	12.22	Friendship building	9–11 weekly sessions lasting 40–50 min each.	Cluster randomization of 14 classes across four schools.	Combination of two interventions (RAP and PIR) implemented together in a school during class time by trained facilitators, where one targets social network building and the other targets depression prevention.	Students reported significantly higher levels of peer interpersonal relatedness when reassessed 12 months after the intervention. Significant increases in social functioning with peers was also noted.
Wyman et al. (2010), USA [51]	RCT	2675	15–16	Suicide prevention and gate-keeper training	Three phases: (1) school and community preparation, (2) peer leader training, and (3) schoolwide messaging. Phases 1 and 3 ranged in time and phase 2 was 4 h long.	Students recruited from high school. Peer leaders chosen to reflect a diverse range of friendship groups.	Certified trainers focused training on coping skills and available resources such as friends. Trained students (peer leaders) encouraged their friends to reach out to trusted adults and disseminated messages about the intervention through PSAs and social media.	Trained peer leaders were more likely to refer a suicidal friend to an adult and the intervention increased perceptions of adult support in students with a history of suicidal ideation. The intervention was also found to increase the acceptability of seeking help and the confidence to seek and provide support.
Yamaguchi et al. (2020), Japan [41]	Quasi-experimental	899	15–16	MH literacy and support skills intervention	50 min	Participants recruited from a high school.	Teacher-delivered intervention through films, discussions, and role-play.	The intervention showed short-term and long-term (2 months) effects. Students’ intention to seek help and intention to help a peer with signs of mental illness increased.

Note: * Denotes approximate age of participants, as the study provided grade information only.

## Data Availability

All relevant data included in the paper and Appendix A.

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
