# Peer review of "Investigating the Role of Friendship Interventions on the Mental Health Outcomes of Adolescents: A Scoping Review of Range and a Systematic Review of Effectiveness"

_ijerph, 2023, doi:10.3390/ijerph20032160_

Round 1
Reviewer 1 Report
Congratulations on the paper! It is a very important topic, and the literature review includes relevant studies for a topic that is very important in today's world environment. The COVID-19 pandemic gave us valuable lessons and the impact of isolation on adolescents was higher than for other groups. Mental problems are on the rise for all, but adolescents in particular need more programs and support structures. This paper is giving valuable ideas to better address this need. Good job!
Author Response
Thank you very much for your kind review. We agree that we need to broaden the scope of our interventions and how important it is to address social isolation.
Reviewer 2 Report
Dear Authors
Congratulations on a very well researched and crafted article. The approach to both the scoping and systematic reviews is well described and implemented. The themes from the scoping review identified the extent and gaps in research of peer based adolescent friendship mental health interventions. In turn this outcome paved the way for more in depth examination in a systematic review of articles that had evaluated the types of relevant interventions.
The systematic review follows a well recognised approach to systematic reviewing with it being registered with Prospero. Themes are well analysed with excellent flow charts for both reviews and articles well annotated. The final discussion emphasised the value of friendship from both FT and FR perspectives. The overall reviews do raise the sparseness of friendships being used within interventions which is of value to both practitioners and researchers.
My only suggestion for change is to add some introductory comments as a segway between the scoping and the systematic review before going straight into methodology of the latter even if to repeat what was introduced in the last paragraph of the introduction of the article.
Author Response
Thank you for asking us to segue between the scoping and systematic review. We have now added the following to the start of the Systematic Review on page 9:
‘The scoping review allowed us to map the research landscape of existing friendship interventions for adolescent mental health. It illustrated key themes and concepts addressed in the included studies and identified the types of studies which have been conducted. In order to better understand the evidence existing around the effectiveness of the included friendship interventions, a systematic review was then conducted to identify, appraise, and synthesize those studies identified that had available evaluation and post-intervention data (i.e. RCTs; pre-post studies).’
Reviewer 3 Report
This review is very important because it sheds light on the importance of friendship for adolescents.
The title is attractive and very suitable for review
The introduction is well-formatted and contains appropriate, documented, and recent references.
Materials and Methods
It is written systematically
However, the authors did not detail the exclusion criteria in the form of a headline
The systematic review was carried out in an academic and appropriate manner
Strengths and Limitations
Do not generalize the results because the influence of friendship is closely related to customs, traditions, culture and religion
The study was done in developed countries. It is possible to request studies and research in underdeveloped countries
Author Response
Thank you for raising this point about the cultural context of friendships and also the bias towards high-income countries. We have now addressed both these points in the text and in the limitations section on page 22 have, for example, added:
‘Lastly, the included interventions, coming mainly from high-income countries, had limited adaptation for different socio-cultural contexts. Friendship dynamics and values can differ across cultures [70] and future studies should delineate which type of interpersonal interaction is being targeted, if working in a new environment. Identifying components that can be tailored, within interventions, to different contexts will likely better enable interventions to be delivered in new areas. The many apparent gaps in intervention research on mental health will hopefully serve to encourage the development of friendship interventions for adolescent mental health support in the future.’
Reviewer 4 Report
Excellent manuscript with few problems.
Facts were stated correctly and very easy to read and understand/
Author Response
Thank you for taking the time to review our paper and for your generous comments.
Reviewer 5 Report
Thank you so much for giving me the opportunity to review this interesting paper. Overall, it is very well-organized, organized and well-written. I also consider that the topic is relevant. However, there are some small changes that would probably improve this paper.
In particular, I think it would be worth mentioning what is the role of social media and other virtual ways of support?
I think it is a very relevant topic that should be mentioned?
What is the role of the social media in all these processes, could you also include appropriate references?
Author Response
We appreciate your time and effort in reviewing our paper. Thank you so much for your kind words and for raising very important and relevant points about elaborating on the role of social media. We agree that social media and virtual communication/support sources can play a role. We have addressed your comments and included a section on it in our paper on page 20:
‘Recent literature has found online social platforms as beneficial for adolescent friendships, with younger adolescents with access to online platforms at times using online and offline communication with little differentiation [64]. Apart from social media, communication through and during online gaming is also a popular way of online communication and friendship-building among adolescents, as spending time working together on the same task can strengthen friendships [64]. Social media can also serve as a supportive platform as active adolescent users of social media report seeking informal support from friends and peers online, which is why it is important to introduce interventions which involve online friends and understand whether these interventions can play a role in enhancing online friendships [65].’
Round 2
Reviewer 3 Report
Thank you so much
The article is suitable for publication